# OpenReview forum: "How to Evaluate and Mitigate IP Infringement in Visual Generative AI?"
_ICML.cc/2025/Conference — ICML 2025 poster_

### Official Review · Reviewer_1aVv · 2025-02-22

**Overall Recommendation:** 3

**Summary:**

This paper explores the intellectual property (IP) infringement risks posed by state-of-the-art visual generative AI models, such as DALL-E 3, Stable Diffusion XL. The study shows that these models can generate content resembling IP-protected characters (e.g., Spider-Man, Iron Man, Superman) even when given prompts that do not explicitly mention their names. The authors develop a benchmarking framework to evaluate infringement and propose a mitigation strategy TRIM that detects and suppresses infringing outputs using guidance techniques in diffusion models.

**Claims And Evidence:**

Yes.

**Essential References Not Discussed:**

None.

**Experimental Designs Or Analyses:**

The experiments are well-structured.

**Methods And Evaluation Criteria:**

The benchmarking approach is well-designed and directly relevant to the problem.

**Other Comments Or Suggestions:**

None.

**Other Strengths And Weaknesses:**

**Strengths:**
This paper introduces a well-structured benchmarking framework to systematically evaluate IP infringement across multiple AI models and characters.
The proposed TRIM method effectively prevents IP infringement without requiring model retraining.

**Weaknesses:**

The proposed method depends on predefined lists of IP-protected characters. How would the system adapt to new characters or lesser-known copyrighted content?

Any existing mitigation techniques can be compared with TRIM?

This paper relies on human evaluation to measure IP infringement in the generated content. While the authors mention that the human evaluators are familiar with the characters involved, they do not provide sufficient details about the evaluators' backgrounds or their familiarity with intellectual property (IP) law. Assessing IP infringement is not just about visual similarity; it also involves legal judgments about whether the generated content constitutes a violation of copyright.

The paper provides only one visual example (Figure 12) comparing the generated images before and after applying the proposed TRIM method. While this example effectively demonstrates the mitigation of IP infringement for the character Spider-Man, it is insufficient to fully evaluate the generalizability and effectiveness of the method across different characters and scenarios. The authors should include more visual examples of the generated content, especially for other well-known IP-protected characters like Iron Man, Superman, and Batman, as well as non-human characters. Additionally, visual examples of failed cases or edge cases (e.g., where TRIM fails to mitigate infringement or where it overly suppresses the generation of non-infringing content) would provide a more comprehensive understanding of the method's strengths and limitations.

**Questions For Authors:**

see Other Strengths And Weaknesses.

**Relation To Broader Scientific Literature:**

The study is highly relevant to AI ethics, fair use in AI-generated content, and legal compliance in generative models.

**Theoretical Claims:**

No formal proofs are presented.

---

> ### Author Rebuttal · Authors · 2025-04-01
>
> Thank you very much for your insightful comments. We hope the following results
> and clarifications can address your concerns. Please let us know if anything is
> still unclear. We are more than willing to provide further clarification and
> conduct more experiments if needed.
>
> **Q1:** The proposed method depends on predefined lists of IP-protected
> characters. How would the system adapt to new characters or lesser-known
> copyrighted content?
>
> **A1:** Thanks for your thoughtful question. In this paper, we focus on
> defending against IP infringement involving well-known, existing intellectual
> property (IP) content, as these characters are typically owned by major
> entertainment companies and are associated with significant financial value.
> Thus, these IP content can be listed in a predefined list. Since IP infringement
> by visual generative AI often stems from memorization of training data, it is
> less likely for text-to-image and text-to-video models to reproduce infringing
> content related to newly created or lesser-known copyrighted works. As a future
> direction, we aim to make the system adaptable to such content without
> retraining large multimodal language models used in the detection process, for example through knowledge
> editing techniques like those proposed by Cheng et al.
>
> Cheng et al., Can We Edit Multimodal Large Language Models? EMNLP 2023.
>
> **Q2:** Any existing mitigation techniques can be compared with TRIM?
>
> **A2:** Thank you very much for your insightful question. Please refer to
> Reviewer-uqQi-A3 and Reviewer-uqQi-A4.
>
> **Q3:** This paper relies on human evaluation to measure IP infringement in the
> generated content. While the authors mention that the human evaluators are
> familiar with the characters involved, they do not provide sufficient details
> about the evaluators' backgrounds or their familiarity with intellectual
> property (IP) law. Assessing IP infringement is not just about visual
> similarity; it also involves legal judgments about whether the generated content
> constitutes a violation of copyright.
>
> **A3:** Thanks for your insightful comments. We ensure our annotators are
> familiar with the judgment process in the real-world related lawsuit case
> Andersen v. Stability AI Ltd., 23-cv-00201-WHO. Thus, we believe our annotators
> can be considered as legally knowledgeable human annotators. We acknowledge that
> the annotators are not experts in intellectual property infringement law (e.g.,
> they do not hold advanced degrees such as a doctoral degree in this field), and
> we will clarify it in the limitation section. We will make it more clear in the
> revised version.
>
> **Q4:** The paper provides only one visual example (Figure 12) comparing the
> generated images before and after applying the proposed TRIM method. While this
> example effectively demonstrates the mitigation of IP infringement for the
> character Spider-Man, it is insufficient to fully evaluate the generalizability
> and effectiveness of the method across different characters and scenarios. The
> authors should include more visual examples of the generated content, especially
> for other well-known IP-protected characters like Iron Man, Superman, and
> Batman, as well as non-human characters. Additionally, visual examples of failed
> cases or edge cases (e.g., where TRIM fails to mitigate infringement or where it
> overly suppresses the generation of non-infringing content) would provide a more
> comprehensive understanding of the method's strengths and limitations.
>
> **A4:** Thank you very much for your constructive commments and valuable
> suggestions. Please see
> https://anonymous.4open.science/r/GAI_IP_Infringement_submission-0A27/more_visualizations.pdf
> for the added visualizations. We have added more visual examples of the
> generated samples by default model and our method on different characters and
> contents (i.e., Spider-Man, Iron Man, Incredible Hulk, Superman, Batman, and
> Coca-cola). We also added the examples of the failure cases accordingly. Will
> add more visual examples in our revised paper.

---

> > ### Comment · Reviewer_1aVv · 2025-04-02
> >
> > Thank you for the clarification. I will consider this rebuttal result in my final decision.

---

### Official Review · Reviewer_QAWe · 2025-03-05

**Overall Recommendation:** 4

**Summary:**

The paper presents a method for creating prompts that may cause T2I/T2V models to generate images infringing on IP rights and show that IP infringement issues are widespread across different visual generative models based on their constructed
prompts. They then develop a defensive method which combines detecting IP infringing contents using LLM and VLM and suppress IP infringement by modifying the diffusion generation process. Experiments show that the designed defense can effectively reduce the IP infringement in visual generative models.

**Claims And Evidence:**

This is an empirical paper and the claims in this paper are well supported by the experiments.

**Essential References Not Discussed:**

It is suggested to add the discussion about the transferability of the proposed defense to other model architectures besides diffusion, such as autoregressive models [1].

**Experimental Designs Or Analyses:**

The experiment designs and the results are clear. I notice that the IP infringement rates for different characters is different. For example, the average infringement rates for Spider Man is larger than the rates for Iron Man. Could the authors provide some analysis to this phenomenon?

**Methods And Evaluation Criteria:**

The weaknesses of the proposed method:

* As the proposed defense works by modifying and controlling the diffusion process of the model, this method might require the white-box access of the T2I/T2V models. Its application on the models with only black-box access might be limited.

* The proposed defense method is specifically designed for diffusion-based Text-to-Image (T2I) and Text-to-Video (T2V) models. Its effectiveness may not extend to other architectures such as GANs or autoregressive models like VAR [1]. This limitation exists because the defense operates by suppressing intellectual property infringement through targeted modifications to the diffusion process itself. Consequently, its transferability to models with fundamentally different architectures remains uncertain.

* The revised generation process in the proposed defense could introduce large runtime overheads.

[1] Visual Autoregressive Modeling: Scalable Image Generation via Next-Scale Prediction. NeurIPS 2024.

Except the above weaknesses, the methods and the evaluation criteria in this paper are easy-to-understand and sound.

**Other Comments Or Suggestions:**

See above

**Other Strengths And Weaknesses:**

Strengths:

* The topic is important and trendy. The findings and the contributions in this paper could be significant for both the visual generative AI field and and the field of AI safety and governance.

* The method for evaluating the IP infringement in T2I/T2V models is new, and it's a general method to test the IP infringement for different contents.

* The performance of the proposed defense method is promising.

* This paper is well-structured and easy-to-follow.

Weaknesses:

* The proposed defense method might be specifically designed for diffusion-based Text-to-Image (T2I) and Text-to-Video (T2V) models.

* The proposed defense method requires the white-box access to the models.

* The revised generation process in the proposed defense could introduce large runtime overheads.

* There are various symbols used in this paper, especially Algorithm 1. It is suggested to have a table to summarize the meaning of different symbols.

Considering the strengths and the weaknesses of this paper, I lean to accept.

**Questions For Authors:**

No.

**Relation To Broader Scientific Literature:**

The key contributions of this paper can be summarized as: 1. Proposing an approach to evaluate the IP infringement on T2I/T2V models. 2. Showing that the IP infringement issues on recent T2I/T2V models are common. 3. Proposing an
effective defense method for IP infringement on T2I/T2V models. These key contributions are related to the existing research in both the visual generative AI domain (such as the diffusion model community) and the field of AI safety and governance.

**Theoretical Claims:**

This paper is mainly empirical based and it does not include theoretical claim and proof.

---

> ### Author Rebuttal · Authors · 2025-04-01
>
> Thank you very much for your valuable comments. We hope the following results
> and clarifications can address your concerns. Please let us know if anything is
> still unclear. We are more than willing to provide further clarification and
> conduct more experiments if needed.
>
> **Q1:** As the proposed defense works by modifying and controlling the diffusion
> process of the model, this method might require the white-box access of the
> T2I/T2V models. Its application on the models with only black-box access might
> be limited.
>
> **A1:** Thank you for your insightful comments. In this paper, we focus on the
> threat model where the defender has white-box access to the model—a practical
> assumption, as IP infringement concerns typically arise for model owners
> themselves. For example, companies like OpenAI and Midjourney Inc. are likely to
> protect their web or API-based services (e.g., DALL-E 3, Midjourney) from IP
> violations. With white-box access, they can directly integrate our method into
> their generation pipelines. Defense under the threat model where the defender
> only has the black-box access of the model is out of the scope of this paper and
> will be our future work. We will clarify this point further in the revised
> version.
>
> **Q2:** Transferability to autoregressive models.
>
> **A2:** Thank you very much for your thoughtful comment. In this paper, we focus
> on the defense for diffusion-based visual generative models. Yao et al. have
> shown that diffusion-based models outperform purely autoregressive model VAR in
> image generation. Specifically, many diffusion models (e.g., MDTv2, REPA, and
> LightningDiT) achieve better performance than VAR. Most state-of-the-art
> open-source and proprietary visual generative models—such as Midjourney, DALL-E,
> Flux, and Ideogram—are based on diffusion. Even for recent GPT-4o image
> generation, it's also possible that it is implemented by the combination of
> autoregressive and diffusion. Developing defense methods for autoregressive
> models is an important direction for future work. We will clarify further in the
> revised version of our paper.
>
> Yao et al., Reconstruction vs. Generation: Taming Optimization Dilemma in Latent Diffusion Models. CVPR 2025.
>
> **Q3:** The revised generation process in the proposed defense could introduce
> large runtime overheads.
>
> **A3:** Thanks for your helpful questions. Table 7 in the Appendix reports the
> average runtime of each process on Stable Diffusion XL. Image generation with
> classifier-free guidance on detected IP names takes nearly the same time as
> standard generation (32.25s vs. 32.04s). Character name detection (0.42s) and
> image infringement detection (2.42s) add minimal overhead. For benign images,
> the total added cost is just 0.21s. For infringing images, the runtime roughly
> doubles due to a second diffusion step. However, since only a small fraction of
> images (those detected with IP infringement issues) proceed to the second
> diffusion process (lines 10–14 in Algorithm 1), and this step effectively
> mitigates IP infringement, we believe the runtime cost of our method to be
> acceptable. We will include a more detailed discussion in the revised version of
> our paper.
>
> **Q4:** I notice that the IP infringement rates for different characters is
> different. For example, the average infringement rates for Spider Man is larger
> than the rates for Iron Man. Could the authors provide some analysis to this
> phenomenon?
>
> **A4:** Thanks for your insightful question. Since visual generative models are
> trained in a data-driven manner, potential IP infringement by these models can
> often be traced back to memorization of the training data. Studies by Somepalli
> et al. and Carlini et al. have shown that the degree of memorization is
> correlated with the number of duplicate samples in the training set. As a
> result, characters with more training samples tend to have higher IP
> infringement rates. We will include a more detailed analysis of this phenomenon
> in the revised version of our paper.
>
> Somepalli et al., Understanding and Mitigating Copying in Diffusion Models. NeurIPS 2023.
>
> Carlini et al., Extracting Training Data from Diffusion Models. USENIX Security 2023.
>
> **Q5:** There are various symbols used in this paper, especially Algorithm 1. It
> is suggested to have a table to summarize the meaning of different symbols.
>
> **A5:** Thank you very much for your helpful suggestion. We will add the table
> for summarizing the meaning of different symbols accordingly in our revised
> version.

---

### Official Review · Reviewer_12AN · 2025-03-13

**Overall Recommendation:** 4

**Summary:**

This paper illustrates that IP Infringement often happens under both ‘Name-based Lure Prompt’ and ‘Description-based Lure Prompt’ situations. Then, this paper proposes a defensive method, named TRIM, to mitigate the infringement. TRIM blocks the targeted name and detect the infringement to regenerate the image. This paper investigates an interesting problem: how to evaluate copyright infringement in text-to-image and text-to-video models, how severe these copyright issues are for current state-of-the-art models, and how to mitigate
such issues. It proposes methods to construct "lure prompts" for testing copyright infringement severity and also introduces techniques to mitigate copyright infringement during generation time. Based on the experimental results, current visual generative models have severe copyright infringement issues, and the proposed mitigation framework can effectively reduce the probability of copyright infringement.

This is a well-written paper. The studied problem in this paper is highly practical, timely and impactful. The proposed "lure prompts" based copyright infringement evaluation is novel. The proposed mitigation framework is simple yet effective. There are some concerns regarding the evaluation especially the results under adversarial jailbreak attacks.

**Claims And Evidence:**

The IP Infringement problem is supported by clear and convincing evidence from the experiment and references. The main claims in this paper, such as the severe copyright issues associated with the current state-of-the-art models and the effectiveness of designed mitigation are all well supported by the evidence in the experiment results.

**Essential References Not Discussed:**

*Essential References Not Discussed Kim et al. proposes a jailbreak attacks forcing text-to-images models to generate copyright infringing content. The evaluation of the proposed mitigation method under this attack is unclear.

Kim et al., Automatic Jailbreaking of the Text-to-Image Generative AI Systems.

**Experimental Designs Or Analyses:**

Overall, the experiment results in this paper is sufficient to support the main claims of this paper. The weakness of the experimental designs is that the evaluation under adversarial settings for the proposed mitigation method is missing. For example, there are some adversarial jailbreak method for forcing text-to-image models to generate copyright infringing images, such as [A], The evaluation of the robustness of the proposed mitigation method against such attack might be helpful. The criterion for human judgment should be listed in the experiment settings. It’s better to explain why CLIP score is suitable to evaluate the performance of the proposed method.

[A] Kim et al., Automatic Jailbreaking of the Text-to-Image Generative AI Systems.

**Methods And Evaluation Criteria:**

In general, the proposed method to construct the "lure prompts" and evaluate the copyright infringement in text-to-image and text-to-video models are reasonable. The proposed mitigation framework also makes sense, and shows effectiveness with acceptable efficiency overhead based on the results. The reason for the selection of the evaluation criteria is clearly discussed in Section 4. Regarding the evaluation criteria for the proposed mitigation method, since it initially employs a VLM to detect copyright infringement in generated images, it would be beneficial to measure the recall rate of this detection process. More explanation might be needed:
1. The weight for the U-Net needs more explanation. Currently, authors only claim that the value is 7.5 without further explanation.
2. Some other notations also need to be explained, e.g. \tilde{\epsilon}.

**Other Comments Or Suggestions:**

n/a

**Other Strengths And Weaknesses:**

n/a

**Questions For Authors:**

1. What is the recall rate of this VLM-based detection process in the proposed mitigation method?
2. What is the robustness of the proposed mitigation method under jailbreak attacks targeting on copyright infringement?
3. Does the evaluation for the experimental results refer to utilising techniques such as crowdsourcing?

**Relation To Broader Scientific Literature:**

The contribution of this paper is suitable for the scope of IP Infringement, and it could have large impact to the text-to-visual-content community.

**Theoretical Claims:**

N/A

---

> ### Author Rebuttal · Authors · 2025-04-01
>
> Thank you very much for your helpful comments. We hope the following results
> and clarifications can address your concerns. Please let us know if anything is
> still unclear. We are more than willing to provide further clarification and
> conduct more experiments if needed.
>
> **Q1:** Recall rate of VLM-based detection process.
>
> **A1:** Thank you very much for your constructive question. The average
> detection recall rates on differernt visual generative models are shown as
> follows:
>
> Character | Recall
> ---- | ---
> Spider Man | 0.98
> Iron Man | 0.99
> Incredible Hulk | 0.99
> Super Mario | 1.00
> Batman | 0.96
> SuperMan | 0.98
>
> As can be observed, the VLM-based detection has high recall rates for detecting
> IP infrigement. We will add more results in our revised version.
>
> **Q2:** The weight for the U-Net needs more explanation. Currently, authors only
> claim that the value is 7.5 without further explanation.
>
> **A2:** Thanks for your thoughtful comment. The weight applied to the U-Net
> controls the trade-off between suppressing IP infringement and maintaining
> generation quality. A higher weight leads to stronger suppression, but it may
> also degrade the quality of the generated images (such as the text-image
> alignment). Wang et al. found that a value of 7.5 provides a good balance in
> most classifier-free diffusion guidance settings. Therefore, we adopt 7.5 as the
> default value in our setup. We will clarify this point in the revised version of
> our paper.
>
> Wang et al., Analysis of Classifier-Free Guidance Weight Schedulers. TMLR 2024.
>
> **Q3:** Some other notations also need to be explained, e.g. \tilde{\epsilon}.
>
> **A3:** Thank you very much for your useful comment. \tilde{\epsilon} means the
> mapping between the predicted noise and the input noise, prompt as well as the
> timestep in the revised diffusion process. Will make it more clear in the
> revised version.
>
> **Q4:** Robustness of the proposed mitigation method under jailbreak attacks
> targeting on copyright infringement.
>
> **A4:** Thank you very much for your constructive questions. we conducted the
> suggested experiments to evaluate our defense against the adversarial
> infringement method proposed by Kim et al. [A], using Stable Diffusion XL. The
> results on Spider-Man are as follows:
>
> Method | IP Infringement Rate
> ---- | ---
> Undefended | 81.6%
> TRIM (Ours) | 6.8%
>
> The results on Superman are as follows:
>
> Method | IP Infringement Rate
> ---- | ---
> Undefended | 94.6%
> TRIM (Ours) | 8.4%
>
> Our method shows strong robustness against adversarial infringement, as the
> classifier-free guidance mechanism effectively constrains the output space,
> preventing alignment with protected IP. We will include additional experiments
> and discussions in the revised version. Thank you again for your valuable
> feedback.
>
> **Q5:** The criterion for human judgment should be listed in the experiment
> settings.
>
> **A5:** Thanks for your useful question. In our evaluation, the annotators are
> familiar with the judgment process used in the real-world lawsuit Andersen v.
> Stability AI Ltd., 23-cv-00201-WHO. They are instructed to assess whether an
> image constitutes IP infringement by selecting either "yes" or "no," based on
> their understanding of the legal reasoning and criteria outlined in that case.
> We will make it more clear in our revised version.
>
> **Q6:** It’s better to explain why CLIP score is suitable to evaluate the
> performance of the proposed method.
>
> **A6:** Thanks for your thoughtful comment. The CLIP score is used to evaluate
> text-image alignment in visual generative models, which is a key quality metric
> for text-to-image generation. Empirical studies, such as Hessel et al., have
> shown that the CLIP score correlates strongly with human judgments of how well
> an image matches its corresponding caption. Will make it more clear.
>
> Hessel et al., CLIPScore: A Reference-free Evaluation Metric for Image Captioning. EMNLP 2021.
>
> **Q7:** Does the evaluation for the experimental results refer to utilising
> techniques such as crowdsourcing?
>
> **A7:** Thank you very much for your constructive question. Yes, the evaluation
> of our experimental results is based on a form of crowdsourcing. We will make it
> more clear.

---

### Official Review · Reviewer_uqQi · 2025-03-13

**Overall Recommendation:** 4

**Summary:**

This paper discovers that SOTA diffusion models tend to generate content that very highly resembles data that could be protected by IP rights, for example, Marvel characters. Their human evaluation studies show that the risk with these characters/concepts is very high. They also propose a mitigation method that uses the names/descriptions of these characters as a negative prompt in classifier free guidance.

**Claims And Evidence:**

The paper claims to recognize protected content and mitigate copyright infringement by modifying the generation process. Although their method significantly improves upon existing models, I believe the corpus of selected characters/ content is quite limited. These models have been shown to memorize many images within the training data and the style of certain artists. But the authors do not test their model on these concepts. Therefore, the claims may be over stated.

**Essential References Not Discussed:**

I believe some papers like [1, 2], related to memorization in diffusion models, have not been discussed.

[1] Finding NeMo: Localizing Neurons Responsible For Memorization in Diffusion Models, Hintersdorf et al
[2] Memorized Images in Diffusion Models share a Subspace that can be Located and Deleted, Chavhan et al

**Experimental Designs Or Analyses:**

I think the experimental design and analysis are mostly sound. However, I would like to highlight the following -

The proposed method will fail in a white-box adversarial attack as the adversary can simply disable the infringement detection model. Many other memorization papers in diffusion models like [1,2,3,4] have the same problem. Can the authors compare their work with these papers?
[1] Understanding and Mitigating Copying in Diffusion Models, Somepalli et al
[2] Exploring Local Memorization in Diffusion Models via Bright Ending Attention, Chen et al
[3] Unveiling and Mitigating Memorization in Text-to-image Diffusion Models through Cross Attention, Ren et al
[4] Finding NeMo: Localizing Neurons Responsible For Memorization in Diffusion Models, Hintersdorf et al

Concept Erasure in diffusion models can erase the knowledge of the objects from the model itself. These models can still generate safe images under white box attacks as the model parameters have been changed to forget undesired concepts. Is the proposed better than concept erasure methods? How do the authors propose to handle white-box attacks?
[5] Erasing Concepts from Diffusion Models, Gandikota et al
[6] Unified Concept Editing in Diffusion Models,  Gandikota et al
[7] ConceptPrune: Concept Editing in Diffusion Models via Skilled Neuron Pruning, Chavhan et al
[8] Ablating Concepts in Text-to-Image Diffusion Models, Kumari et al

**Methods And Evaluation Criteria:**

The application at hand is mitigating the generation of copyrighted content in SOTA diffusion models. The methods and the evaluation criteria make sense in the context of the problem considered.

**Other Comments Or Suggestions:**

I believe this paper could have a significantly stronger impact if Groot was added. Because, I am Groot.

**Other Strengths And Weaknesses:**

Strengths -
1. The paper is well written and the methodology is sound.
2. Their method is very effective in protecting copyrighted data.

Please see Experimental Designs or Analyses section for Weaknesses.

**Questions For Authors:**

Please see Experimental Designs or Analyses section. I have some extra questions -

1. How did the users in your human evaluation detect infringement? More specifically, where they given a 'yes' or 'no' option or a Likert scale?
2. Would it be useful to have an LLM perturb the prompt in a way such that the model generates a different image? What are your thoughts on this?

**Relation To Broader Scientific Literature:**

The paper contributes to mitigating memorization in diffusion models, which is an upcoming field. The key contributions of this paper are - using an infringement detection module in the diffusion model pipeline, and using the 'protected' concept as negative prompt for classifier-free guidance. I believe the method proposed in this paper is related to other papers in this field where they propose some form of prompt perturbation.

**Theoretical Claims:**

No theoretical claims are made in this paper.

---

> ### Author Rebuttal · Authors · 2025-04-01
>
> Thank you very much for your thoughtful comments.
>
> **Q1:** Corpus of selected characters/content.
>
> **A1:** Thank you very much for your constructive comments. Besides the results
> in our main paper, we also have the results on different types of non-human IP
> contents in Table 6 in the Appendix. We also conducted more experiment to
> include more IP content. The results alongside the characters in our main paper
> are as follows:
>
> IP | Method | IP Infringement Rate
> ---- |---- | ---
> Mercedes-Benz |Undefended | 42.4%
> Mercedes-Benz |TRIM (Ours) | 0.0%
> Nike |Undefended | 33.6%
> Nike |TRIM (Ours) | 0.8%
> Coca-Cola |Undefended | 76.4%
> Coca-Cola |TRIM (Ours) | 2.0%
>
> The model used here is Stable Diffusion XL. The results demonstrate that our
> defense method is generalizable to different media formats and different IP
> content types on visual generative AI. We also conducted additional experiments
> using name-based lure prompts to evaluate IP infringement on van Gogh and Ghibli
> styles using the recent GPT-4o model, and on the character "Groot" in Stable
> Diffusion XL and Stable
> Diffusion XL Turbo. The results are as follows:
>
> IP | Model | IP Infringement Rate
> ---- |---- | ---
> van Gogh style |GPT-4o | 97.2%
> Ghibli style |GPT-4o | 98.6%
> Groot |Stable Diffusion XL | 99.0%
> Groot |Stable Diffusion XL Turbo| 100.0%
>
> We will add more results in our revised version.
>
> **Q2:** How do the authors propose to handle white-box attacks?
>
> **A2:** This paper focuses on a threat model where the defender has white-box
> access to the model, while the attacker only has black-box access—such as
> through a website or API. This is a practical setting for proprietary,
> closed-source models like DALL-E 3 and Midjourney, which represent most
> state-of-the-art models today. Defending open-source models against white-box
> attacks is beyond the scope of this work and will be explored in future
> research.
>
> **Q3:** Comparison to memorization mitigation methods.
>
> **A3:** We'd like to clarify that our work addresses a broader issue than
> memorization mitigation. While memorization papers focus on preventing models
> from reproducing nearly exact training images, we target IP infringement —
> including outputs that resemble copyrighted content even without nearly exact
> matches. What these papers considers successful mitigation often still qualifies
> as infringement under our evaluation. For example, in Figure 3 of Hintersdorf et
> al., the generated image no longer matches training samples but still clearly
> violates the IP of DC's "Hawkgirl." We also compare our method and the
> open-sourced inference-time memorization mitigation approaches suggested using
> Stable Diffusion XL and Spider-Man. The results are as follows:
>
> Method | IP Infringement Rate
> ---- | ---
> Undefended | 76.6%
> Somepalli et al. | 69.4%
> Ren et al . | 30.2%
> Hintersdorf et al. | 43.2%
> TRIM (Ours) | 5.8%
>
> The results demonstrate that our method is more effecive for mitigating the IP
> infringement. We will add more discussion in our revised version.
>
> **Q4:** Comparison to concept erasure methods.
>
> **A4:** Existing concept erasure methods often degrade the quality of all
> outputs, including non-infringing ones. For example, on Stable Diffusion,
> removing "Spider-Man" with Gandikota et al. [6] results in an LPIPS of 0.23;
> erasing 100 concepts with UCE [7] leads to 0.30 LPIPS; ConceptPrune [7]
> increases FID by 16.6% when removing 5 artist styles; and Kumari et al.'s method
> reduces CLIP scores by ~5% on non-infringing images. In contrast, our method
> does not influence the quality of non-infringing outputs, which represent the
> majority of generated content. These methods are also computationally expensive
> (e.g., ~170 minutes per concept in [6]) and less effective (e.g., 11.2%
> Spider-Man infringement rate [6] vs. 0.0% with ours). Moreover, their
> performance worsens as more concepts are erased. Our method handles multiple IPs
> efficiently without influencing output quality. We will elaborate further in the
> revised version.
>
> **Q5:** I believe some papers like [1, 2], related to memorization in diffusion
> models, have not been discussed.
>
> **A5:** Thank you very much for your helpful suggestion. We will add
> more discussion on the suggested papers related to memorization in
> diffusion models accordingly.
>
> **Q6:** How did the users in your human evaluation detect infringement? More
> specifically, where they given a 'yes' or 'no' option or a Likert scale?
>
> **A6:** In our evaluation, the annotator gives
> a 'yes' or 'no' option. We will make it more clear in our revised version.
>
> **Q7:** Using an LLM to perturb the prompt.
>
> **A7:** We conducted experiments using GPT-4o to rewrite and perturb prompts to
> test its ability to reduce IP infringement. For Spider-Man with Stable Diffusion
> XL, the infringement rate dropped by only 2.7%. This shows that simply using an
> LLM to change the prompt is not effective, since the key semantics and meaning
> in the prompt that leads to IP infringement is still there.

---

> > ### Comment · Reviewer_uqQi · 2025-04-07
> >
> > Thank you for your rebuttal. I will increase my score to 4.

---

### Decision · Program_Chairs · 2025-05-01

**Decision:**

Accept (poster)

**Comment:**

This paper addresses an important topic in responsive generative AI. The rebuttal successfully addresses the reviewers' questions, leading two of them to increase their ratings. Ultimately, all four reviewers support accepting the submission. The AC agrees with the consensus and recommends incorporating the reviewers' suggestions and addressing the points raised in the ethics review in the final revision.